# A Cross-Sectional Survey of Salty Snack Consumption among Serbian Urban-Living Students and Their Contribution to Salt Intake

**DOI:** 10.3390/nu12113290

**Published:** 2020-10-27

**Authors:** Jasmina B. Timic, Jelena Kotur-Stevuljevic, Heiner Boeing, Dušanka Krajnovic, Brizita Djordjevic, Sladjana Sobajic

**Affiliations:** 1Department of Bromatology, Faculty of Pharmacy, University of Belgrade, Vojvode Stepe 450, 11221 Belgrade, Serbia; brizitadjordjevic@gmail.com (B.D.); sladjana.sobajic@pharmacy.bg.ac.rs (S.S.); 2Department of Medical Biochemistry, Faculty of Pharmacy, University of Belgrade, Vojvode Stepe 450, 11221 Belgrade, Serbia; jkotur@pharmacy.bg.ac.rs; 3Data Analysis Unit, National Institute of Gastroenterology “Saverio de Bellis”, Research Hospital, Castellana Grotte, 70013 Bari, Italy; boeing@dife.de; 4Department of Social Pharmacy and Pharmaceutical Legislation, Faculty of Pharmacy, University of Belgrade, Vojvode Stepe 450, 11221 Belgrade, Serbia; dusica.krajnovic@pharmacy.bg.ac.rs

**Keywords:** salty snack products, students, consumption, salt intake

## Abstract

This study investigated the behavior of urban-living students related to the salty snacks consumption, and their contribution to salt daily intake. A cross-sectional survey on 1313 urban-living students (16–25 years, 61.4% university students and 38.6% high school students) used a pre-verified questionnaire created specifically for the study. The logistic regression analysis was performed to investigate the factors influencing snack consumption. The results of salt content and the snack consumption frequency were used to evaluate snack contribution to salt intake. All subjects consumed salty snacks, on average several times per week, more often at home and slightly more during periods of intensive studying, with 42% of the participants reporting to consume two or more packages per snacking occasion. Most of the participants consumed such products between main meals, but 10% of them took snacks immediately after the main meal. More high-school students than university students were in the “high snack group” (*p* < 0.05). The most frequently consumed salty snacks were those with the highest content of salt. Salt intake from snack products for a majority of participants ranged between 0.4 and 1 g/day. The research revealed younger age, home environment and significant contribution to salt intake as critical points in salty snack consumption among urban-living students important for the better understanding of their dietary habits.

## 1. Introduction

Adolescence is an important period in life in which food choices are starting to become individually settled. Some food choices such as skipping meals, irregular meals, a low consumption of fruits and vegetables, often snacking on high-energy-dense foods, and drinking high-energy-dense beverages, are considered detrimental to health in the long-term [1]. Snacking among children and adolescents has become very common worldwide, with an increase of over 20% on a daily basis over the past 30 years [2]. The attitudes and habits associated with snacking and snack product consumption in the population of young people have been investigated in many countries in recent years [3,4,5,6]. Snacking trends among children in the United States (USA) in the period between 1989 and 2006 saw an enormous increase, with an average of three snacks per day, and more than 27% of daily calories coming from snacks. The highest increases were associated with salty snacks and candies. Over 24% of adolescents in Mexico consumed salty snacks every day [4], whereas young adults aged 16–24 years in United Kingdom consumed snacks more than 2.5 times a day [6]. Apart from the snack consumption frequency and demographic characteristics, it was of interest to study the correlation between snacking and the body mass index, social, economic, and psychological determinants, and the influence of the environment and circumstances on snacking patterns [7,8].

Generally speaking, snack food is a broad term covering a heterogeneous, wide, and diverse range of products, coming in a variety of forms, including pre-packaged snack foods and food items made at home. Most of the studies considered snack foods to be one product group [3,9,10] while researchers less frequently conducted studies that addressed one particular snack type. Salty snacks differ in their composition from other snack products, especially in terms of salt and sugar content [11]. In addition, not much is known to what extent salty snacks contribute to salt intake. The salt content in these food items may be high contributing to an excessive intake of sodium, which is associated with high blood pressure and adverse cardiovascular events. In one of the rare studies of this kind Ponzo et al. [12] have found that almost half of the average daily sodium intake in Italian adolescents originated from salty snacks and that their blood pressure was positively correlated with the frequency of salty snack consumption.

The World Health Organization’s framework "Shake the Salt Habit" emphasizes the importance of identifying the salt dietary sources in order to better formulate salt reduction strategies [13]. Since no previous study of salty snack intake among young people had been conducted in Serbia, the present study investigated high-school and university students from urban areas for the purpose of examining their habits of consuming salty snacks and their importance as salt dietary sources in this population. Our specific objective was to find out whether there are age-related and behavioral differences in the consumption of salty snack products and to identify the factors which contribute to their being high consumers. These kinds of data are also important for developing future educational programs for salt reduction intake and for potential product reformulation options.

## 2. Materials and Methods

### 2.1. Study Population

The final study population included 1313 students from an urban area. The participants were recruited from two public high schools types (*N* = 507; medical school and gymnasium) and two state faculties (*N* = 806; study areas pharmacy and management/organization) in order to portray the attitudes and habits of urban-living young people of different educational background. The schools and faculties were located in the central area of Belgrade, capitol of Serbia, high-school size was >300, and faculty size was >1000 students. Participation was completely voluntary without incentives and required no registration of personal data. The inclusion criterion was age (14–19 for high-school students and 18–27 for university students). The exclusion criterion was the lack of inclusion criteria. Data collection was conducted during 2016 and 2017. The study itself was approved by the Ethics Committee for Biomedical Research of the Faculty of Pharmacy, University of Belgrade, Republic of Serbia (project identification code 164/3, date of approval 25 February 2015).

### 2.2. Questionnaires

Lifestyle and dietary habit questionnaire was compiled specifically for the study and was pre-tested on 30 students prior to the examination to determine the time needed for completion and the level of understanding. The internal consistency of its questions was tested using Cronbach’s α. The survey included 12 brief questions that were divided into several topics and a food frequency questionnaire (FFQ). The participants were asked to fill out the questionnaire in one session in the classroom environment at schools or during breaks between lectures at faculties. All questionnaires were coded with 4-letter codes randomly chosen by the participants. The topics covered by the questionnaire were as follows: demographic characteristics, body weight and height, education level, overall dietary habits, participants’ perception of their diet quality, and their habits related to consuming salty snack products. The self-reported weight and height data of the participants were used to calculate the body mass index (BMI) and further classification of nutritional status was based on the World Health Organization (WHO) criteria [14]. A structured FFQ was additional part of lifestyle and habit questionnaire and was used to collect information about the frequency of salty snack intake (FFQ). In the FFQ, snack products were grouped into categories and included popular salty snack products available on the market, such as chips, flips, popcorn, fried corn, salted sticks, pretzels, crackers, fish-shaped products, roasted and baked nuts, seeds, and salted peanuts. It was also possible to include additional products.

A 72-h dietary recall questionnaire (DR1) on salty snack consumption was conducted on all the participants concurrently with the lifestyle questionnaire, and again 2 months later on 15% of the formerly surveyed population (100 university and 100 high-school students, total *N* = 200). This second survey was not announced in advance and the inclusion criteria were previous engagement in the study and the availability to give answers to DR2 survey (only students that were available and willing at this second occasion participated in DR2). Gender and age characteristics of this sub-group reflected the same characteristics of all participating students. This sub-group of participants was asked to self-code DR2 with the same 4-letter codes as previously. Information about the product type and brand, and the weight and number of packages consumed in last 72 h was collected in DR1 and DR2. The results from DR1 and DR2 for 6 days in total were used to calculate quantitative salt intake (see below).

### 2.3. Salt Content of Snacks

The salt content analysis was carried out employing the Association of Official Analytical Chemists (AOAC) official titrimetric method 950.52 [15]. An internal method control was conducted on a weekly basis using the laboratory chloride standard and samples fortified at 3 concentration levels. A total of 122 samples that included 48 different salty pre-packaged food products were purchased from supermarkets, local shops, and health food stores. The number of samples from different lots that were analyzed ranged between 1 and 7.

### 2.4. Assessment of Chronic Salt Exposure

The results from DR1 and DR2 for 6 days in total for the same sub-group of participants (*N* = 200, confirmed with the match in codes) were used to calculate quantitative salt intake. The observed individual means (OIM) model within the Monte Carlo risk assessment method (MCRA) was employed [16]. In the OIM approach, chronic exposure was calculated using information about salt content in different salty snack products (results obtained from a chemical analysis in our laboratory) and the product type and quantity consumed over the previous 72 h for 200 participants (DR1 and DR2, in total 6 days). Average exposures to salt from salty snack products per participant/day were calculated as follows: each salty snack product consumed by a single participant was multiplied with the mean salt concentration in that product. Exposures from products consumed during 6 recorded days were summed up to obtain the average daily exposure for that person.

### 2.5. Statistical Analysis

Statistical data processing was performed employing Predictive Analytics SoftWare (PASW SPSS), version 18 (Chicago, IL, USA). ANOVA test (with post-hoc Tukey test) for continuous variables comparison, while Chi-square tests for discrete variables were applied to verify the possible differences in sub-groups according to different criteria. For 2 × 2 contingency tables Fisher exact test with Yates correction testing difference in frequencies in distinct categories were implemented.

The logistic regression analysis was performed in order to assess different factors which determined the number of snacks regularly consumed by the subjects of the present study. From the logistic regression models, odds ratios (ORs) were estimated with their corresponding 95% confidence intervals. A *p* value < 0.05 was considered as statistically significant.

Regarding reliability testing of the FFQ’s internal consistency Cronbach’s α coefficient analysis was used. Test–retest analysis was obtained as a correlation analysis between questionnaire’s items in two time points. Exploratory factorial analysis of the questionnaire was made in order to get classified main factors by using principal component analysis as extraction method. Variance value of each factor was measured by distinct eigen value. Factorial analysis validity was tested by the Kaiser–Meyer–Olkin measure of sampling adequacy and Bartlett’s test of sphericity. Rotation was performed with Varimax method. The main characteristic of the extracted factors was percentage of variance explained. Extracted factors were entitled according to items nature and meaning.

## 3. Results

The research covered 1313 respondents, of whom 72.4% were females and 27.6% males. The study population included 806 students from faculties (61.4%) and 507 students from high-schools (38.6%). The respondents’ age ranged from 16 to 25 years. The mean age among students was 23 years and among high-school students 17 years. On average, the mean BMI for university students was 21 ± 2.7 kg/m^2^ and 21 ± 2.5 kg/m^2^ for high-school students. Appendix A features the descriptive characteristics of the population.

Cronbach’s alpha of the questionnaire was 0.865, which is rated as very good reliability. No questionnaire item deleted value was larger than basic value of 0.865, so it could be concluded that all questions were consistent with the questionnaire topic. The interclass correlation coefficient (single measures) was 0.368 (95% confidence interval 0.244–0.535; F = 9.409, df1 = 29, df2 = 2290, *p* < 0.001). Average inter-item correlation coefficient was 0.365 ± 0.039 which suggested relatively strong correlation between items (the average correlation coefficient is acceptable if it is larger than 0.300). Test–retest analysis showed good correlation between different items in two time points (average Spearman’s ρ = 0.785; *p* < 0.01). Principal component analysis method was used for factorial analysis. The Kaiser–Meyer–Olkin parameter for sampling adequacy was 0.608 and the Bartlett’s test of sphericity was statistically significant (*p* < 0.001). Using Varimax rotation four factors were extracted, explaining in total 74.6% of variance. The first factor was related to the main pattern of salty snack consumption (items: dominant place of consumption, frequency of consumption, time of consumption regarding the main meal, and product type) and carried 35.2%; the second that described information influence on snack eating habits (awareness of nutritive value) carried 17,3%; the third 12.3%, and was related to the general dietary habits (items: number of main meals, number of refreshments, eating with nonalcoholic or alcoholic beverages, and eating in front of TV); and the fourth was on personal preferences (university/high school, motivational reasons for consumption, and day period related consumption) carried 9.9% of variance (Appendix A).

As far as the school food environment is concerned, no canteens were available at high schools, but canteens were available for students at their faculties. Vending machines of different types dispensing snacks and beverages were provided on school and faculty premises.

All of the participating students self-reported consuming salty snacks. Serbian youngsters cited “bakery products” (salted sticks, fish-shaped products, pretzels, and bake rolls) as the most popular type of snack products. The next most popularly consumed snacks were chips and popcorn, followed by peanuts, nuts and seeds, flips, and crackers.

The frequency of salty snack consumption, and the number and size of packages consumed were ascertained in the FFQ (Table 1). More than 90% of all the respondents reported eating snack products from several times per month to several times per week. Some 7% of them consumed salty snack products every day, whereas as few as 2.3% reported eating salty snack products most commonly several times a day. Comparing the frequency of snack product intake, substantially different habits were observed among the selected groups. More specifically, the pupil group showed a higher tendency to a more frequent intake of snack products (statistically significant “once per week” and “several times per day”; *p* < 0.05). The results pertaining to the number of packages and time of day when salty snack products were most commonly consumed are featured in Table 1. Concerning the number of packages of salty snack products usually consumed during one snacking occasion, results showed that 37.2% university and 49.3% high-school students regularly consumed multiple packages. More university students than high-school students consumed these products at home (70% and 54%, respectively), while more high-school than university students consumed the same type of snacks at school (34% and 17%, respectively) (Table 1). In terms of the vicinity of the main meals when participants usually consumed salty snack products, the most common answers were “between the main meals” and “immediately after the main meal”. Substantially more high-school students reported that they often take snack products immediately after the main meal and between the main meals as compared to university students (*p* < 0.001), whereas more university students consumed salty snacks instead of the main meal. However, the intake of snacks instead of, before or during the main meal, remained at a very low level. More than one-half of the respondents reported that they sometimes eat snacks in front of the TV set/computer and more than 30% tend to do so very often. The most preferable part of day for salty snack consumption was evening in both groups of students.

The binary logistic regression analysis was performed to assess the factors which contributed to being a high snack consumer (once or more times per day) in comparison with the low group among all subjects of this study and also in high-school/university student subgroups. Statistical significance was taken into account for *p* < 0.05. Resulted significance values (*p*-values) of this part of analyses are presented in Table 3, while in Appendix A a comprehensive statistical data are shown.

Performing the same analysis in separate high-school and university student groups, we revealed that among high-school students the most important predictors of salty snack usage quantity were all three day-period related frequencies, number of refreshments, and self-perception towards diet quality, while in an university student group these factors were midday and evening consumption frequency only. The most important variables (items) that determined the high frequency of salty snack products consumption for all participants together included age, snack type, morning, midday and evening consumption frequency, number of refreshments, and self-perception towards overall diet quality.

Before the assessment of chronic salt exposure was made, the salt content of different salty snacks had been identified. A total of 48 different salty pre-packaged food products were analyzed and the selection of products represented 91% of all the products listed in the questionnaire. The salt content in selected snack products ranged from 0.9 g/100 g (nuts and seeds) to as much as 3.5 g/100 g in Clipsy^®^ product. The analyzed values were in good correlation with the values specified on the product labels (r^2^ = 0.924). The package weight ranged between 18 g and 200 g, depending on the type of salty snacks, and this parameter was of great importance for the potential salt intake in a student population (Table 4).

Evaluation of dietary exposure to salt from various salty snack products consumed by the young population was conducted by analyzing a randomly selected sub-group of 200 participants during two slots of 72 h dietary recall (DR1 and DR2). The average exposure of students to salt from salty snack products was approximately 0.71 g/day (Figure 1). Maximum salt exposure from salty snack products was 2.8 g/day. Salt intake from snack products for the majority of participants ranged between 0.4 and 1 g/day, 33% of all participants were exposed to a daily salt intake below 0.5 g, whereas 18.5% of students showed exposure to salt above 1 g.

## 4. Discussion

This study was the first to explore timing, frequency, and quantity of salty snack consumption and the factors associated with their usage among urban-living high-school and university students in this region and it indicated the notable potential of this food group for increasing daily salt intake. The products that were considered as salty snacks in this study were pre-packaged, available in markets, local shops or traffics near university/high-school, or from vending machines provided on school/university.

### 4.1. Questionnaire Reliability

Reliability analysis regarding internal consistency, so as inter-item correlation showed very good validity of the questionnaire used in this study. Multidimensionality of the questionnaire was analyzed by principal component analysis (PCA). PCA revealed several aspects of this questionnaire: pattern of salty snack consumption, influence of the label information on snack consumption, general dietary habits, and personal preferences which are related to salty snack usage. The four main layers of the questionnaire enabled analyzing snack usage habits of Serbian urban-living students and to make some general conclusions in order to improve eating habits. This also could open the additional exploration possibilities of presented results in PCA extracted factors as separate variables, and their relationship with different study data, but this direction would go beyond the basic aims of the current investigation and is certainly going to be exploited in the future.

### 4.2. General Characteristics of Salty Snack Consumption Patterns

In our study the consumption frequency in urban-living students generally varied from once per weak to several times per week. This is consistent with the results from the northern part of Serbia [17] and north Italy, where the mean weekly consumption of all snacks and salty snacks among students was several servings/week [18]. The findings of the 2016 statistical survey for Europe were similar with the frequency of eating salty snacks at least once a week in 75% of general population in Spain and United Kingdom, and 48% in Germany [19]. Home and evening as preferable location and time of day for salty snacking occasions were common for both student age groups. Similar preferences for Canadian young and adult population were found by Vatanparasat et al. [20]. Higher intake of these food items in the evening in both study groups could be attributed to their having more spare time, spending more time with friends at home or in front of the TV set/computer, or replacing dinner with these products.

Our results enabled testing if there was a correlation between salty snack consumption and the weight of participants. Despite the fact that in Serbia there is an upward trend in obesity among young people as revealed by the statistics of the Institute of Public Health, [21] our subjects from both study groups were of normal body composition (over 90%). Since the weight and height data were self-reported there is a possibility that some under-reporting was present. The “high snack intake group” had slightly lower BMI which could support the hypothesis that higher overall snacking frequency is not associated with the obesity risk in adolescents and young people. The results of other similar investigations of young people’s eating behaviors vary [22,23,24,25]. Although Alphonso et al., found a positive correlation between the BMI and the snacking of chips, popcorn, salted peanuts, and crackers, [26] there was no correlation between the frequency of fast food, soft drink, and candy intake and BMI in the research conducted in the US, where those with a normal BMI consumed 1.1 salty snacks over a period of two days, while overweight, obese, and morbidly obese people consumed 0.9, 1.0, and 0.9 snacks [27]. The review paper of Williamson et al. also indicated that the available evidence do not support snacking as a significant factor leading to weight-related outcomes [28]. No definitive explanation for these findings has been offered, but the reason found in our study could be that the increased energy needs during the period of intensive growth and development were not fulfilled with regular diet, and that the energy contribution from salty snack food items does not exceed the recommended value for particular age. The hypothesis that the quality of overall nutrition may be insufficient to meet the energy needs of young people and consumption of savory snacks is one of the ways to achieve the required energy intake is supported by the fact that over 10% of the students covered by our study consumed salty snacks immediately after the main meal, which could mean that they did not feel satiation after eating. Other possible explanation is that high snack intake increases the feeling of satiety which may lead to skipping main meals and to the lower energy intake.

### 4.3. Factors Associated with High Frequency of Salty Snack Usage

The factors associated with the frequency of salty snack intake in the student urban-living population were investigated. More frequent snacking in all students was associated with day-periods intake, number of refreshments and self-perception towards diet quality (*p* < 0.001). The product type was also an important determinant for a more frequent consumption, which testifies to the relevance of taste factors among young population. Cross et al. [29] 25 years ago concluded that children and adolescents choose snack based on taste over nutrition and more often choose salty, crunchy foods. No correlation was found as regards the number of main meals and the motivation reasons, although some previous studies have found correlation between meal skipping and higher snacking frequency [1]. Additional determinants for increased salty snack frequency consumption in our study were younger age which indicates that younger students can be less prone to the health-conscious eating behavior. The correlation between snacking and gender in our study was not significant and mixed results are available from previous studies [1], although in some of them it was observed that women are often better informed about nutrition and are more inclined to making healthier food choices [30].

As far as the context of salty snacking is concerned, the students included in our study ate less salty snacks at school or faculty and rarely consumed snack food outdoors, which came as a surprise because some previous studies [31,32] revealed that many adolescents consume more snacks while in the company of their peers. The fact that the participants consumed salty snacks mostly at home is also significant for planning possible dietary interventions as it points to the importance of the parental role and the home food environment in replacing the existing dietary patterns with healthier ones. The home was also the most prevalent location for overall snacking in Canadian population [20].

### 4.4. Differences between High School and University Students

The investigated population of high-school students differed from that of university students in several aspects of snack consumption. A substantially higher number of high-school students were in the “high snack intake group” and also more likely to eat multiple packages of salty snacks. There were several reported comparisons between high-school and college participants related to snack usage and salty snack usage, with similar findings of decreasing consumption trend with student maturity [27]. This can be attributed to the greater maturity of university students, or the greater availability of canteens offering main meals at their faculties. A recent study has found that older students, especially in the university setting, are continuously prompted to make healthful food choices, which could also be the reason for a slightly healthier eating behavior [33].

When factors associated with high frequency of salty snack usage were analyzed separately for two student groups certain differences in influencing variables were noticed. Only in high-school student group significant predictor of high salty snack frequency intake was self-perception towards quality of their overall diet, so that better diet quality perception decreased chance for more frequent snack usage. We could assume that perceived poor diet quality in high-school students, at least in part was correlated to their more frequent salty snack use. The awareness of the nutritive value of salty snacks was more important variable for older students’ high usage which was surprising because it was expected that paying attention to the nutritive information on the labels would lead to decreased snack consumption. One of the possible explanations for this finding could be that older students more frequently read the nutritive information but do not understand them properly and do not use them for making more informative food choices and this finding needs further investigation.

### 4.5. Contribution of Salty Snacks to Daily Salt Intake

In terms of potential contribution of salty snacks to the daily intake of salt, the average salt content in different snack types, the package size, and number of packages consumed were all important factors. The most frequently consumed salty snacks were those with the highest salt content (bakery products, flips, clipsy^®^ products, and nuts/seeds). Likewise, taking into account that 40% of participating students consumed multiple snack packages on one snacking occasion, it could be assumed that students may intake significant amounts of salt from snacks. WHO recommends a daily salt intake of <5 g/day for individuals ≥16 years of age (recommended maximum level of intake, (RMLI)) [34]. According to several reports, the intake of salt among the Serbian population is far beyond the recommended amount, reaching 9 g/day and more [35,36] but the contribution of snacks was not previously evaluated. It was of interest to assess the average salt intake from salty snacks on a sample of Serbian urban-living students. The average calculated amount of salt intake from salty snack food represented 14.2% of RMLI, while 18.5% of participants had chronic salt intakes in excess of 20% RMLI only from this food group. If we acknowledge the aforementioned average salt intake for the Serbian population of 9 g/day, then salty snacks contributed an average of 7.9% to the total daily salt intake with a range of 0.22−31.1%. A systematic review of Menyanu et al. [37] identified bread, meat and meat products, sauces, spreads, and different fast foods as the major salt sources in the diets of low- and medium-income countries. Similar findings can be seen in the survey on the Members States’ Implementation of the EU Salt Reduction Framework [38] with bread contributing to the total salt intake with ~20%, meat and meat products with 8–26%, and cheese and dairy products with 10%. If we make a comparison with the relevant data, it is evident that salty snack products could be an equally important food category in the diet of Serbian urban-living students as dairy products or even meat products and consequently they can significantly contribute to their overall salt intake. A recent study by Milincic et al. [39] revealed that 13.1% of adolescents in Serbia had moderately high blood pressure, 5.9% stage 1, and 3.5% stage 2 hypertension, whereas Milosevic Georgiev et al. [40] found elevated systolic blood pressure in 14.9% of Serbian students. Although the correlation between salt intake and hypertension in children and adolescents is still controversial, WHO recommends a reduction of salt intake to control blood pressure in this population [41]. This recommendation was confirmed by the European Salt Action Network in 2018 [42]. Our study indicates that snacking habits potentially represent easily modifiable health risk for urban student population and can be one of the targeted behaviors in adolescence to be dealt with in health management and salt reduction strategies.

### 4.6. Limitations and Implications

The strengths of this study include the range of young people from high school to university students and from different educational fields, and verification of FFQ data with a 72-h recall. The limitations of the study include possible variations in snack salt content, the magnitude of product category covered in the study, the self-reported height and weight, predominant female gender of participants, and a limited number of participants that completed two dietary recalls.

This research focused on snacking habits among young people in a developing country for which there is no sufficient data or applicable national policies and educational strategies. The study results demonstrated that salty snack products may be significant salt contributors for young people. The study important key points are that home environment and younger age population should be the focus of future educational and promotional strategies about healthier food choices. These findings are also useful for defining possible future product reformulation options.

## 5. Conclusions

In conclusion, the findings of this study provide the first information on dietary habits of urban-living students in Serbia connected to salty snacks consumption and their significant contribution to the salt daily intake. Identification of all important dietary sources of salt is in accordance with EU and WHO frameworks for salt reduction and the results of this study could present one important milestone in reduction salt initiatives on national level.

## Figures and Tables

**Figure 1 nutrients-12-03290-f001:**
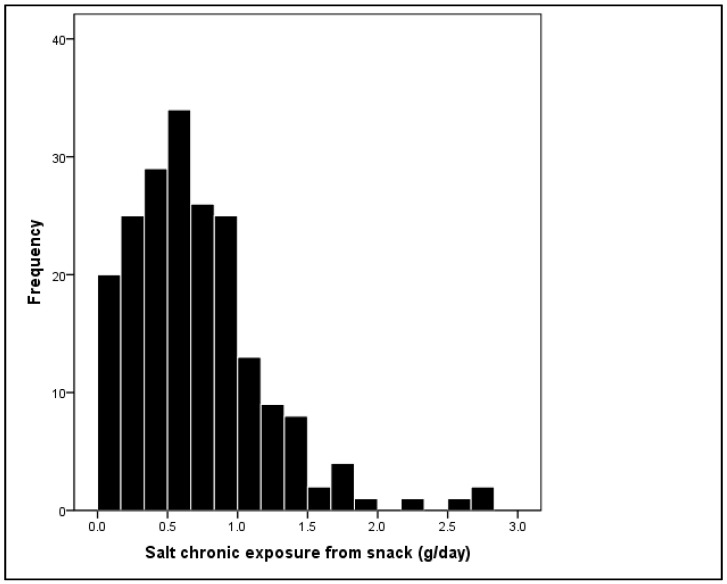
Salt chronic exposure from snack products regularly used among urban student population. *N* = number of participants.

**Table 1 nutrients-12-03290-t001:** Characteristics of salty snack consumption among students.

Frequency of Snack Consumption	All	High-School Student			University Student
1–3 times per month	394 (30.0%)	141 (27.9%)			253 (31.4%)
Once per week	397 (30.2%)	135 (26.7%)			262 (32.5%) *
2–3 times per week	404 (30.8%)	169 (33.4%)			235 (29.1%)
Daily	88 (6.7%)	42 (8.3%)			46 (5.7%)
Several times per day	30 (2.3%)	19 (3.8%)			11 (1.4%) **
χ^2^, *p* * 17.2, <0.01					
Place of usual snack consumption					
At home	819 (62.4%)	262 (51.8%)			557 (69.0%) ***
At school/university	302 (23.0%)	163 (32.2%)			98 (17.2%) ***
Out-of-doors	157 (12.0%)	59 (11.7%)			98 (12.2%)
Other	35 (2.6%)	22 (4.3%)			13 (1.6%) **
χ^2^, *p* * 46.7, <0.001					
Snack consumption in front of the TV or computer					
Often	448 (34.1%)	157 (30.8%)			292 (36.2%)
Sometimes	698 (53.2%)	279 (55.1%)			418 (51.7%)
Never	167 (12.7%)	70 (13.8%)			97 (12.0%)
χ^2^, *p* * 5.7, NS					
Number of snack packages					
One package	763 (58.1%)		256 (50.7%)	507 (62.8%) ***	
Two packages	355 (27.0%)		155 (30.6%)	200 (24.8%) *	
Three packages	135 (10.3%)		61 (12.1%)	74 (9.2%)	
Four packages and more	60 (4.6%)		34 (6.7%)	26 (3.2%) **	
χ^2^, *p* * 26.1, <0.001					
Snack consumption time					
Before the main meal	27 (2.1%)	10 (2.0%)			17 (2.1%)
During the main meal	4 (0.3%)	1 (0.2%)			3 (0.4%)
Immediately after the main meal	116 (8.8%)	55 (10.9%)			61 (7.6%)
Between the main meals	814 (62%)	371 (73.3%)			443 (54.9%) ***
Instead of the main meal	22 (1.7%)	3 (0.6%)			19 (2.4%) *
Other	330 (25.1%)	264 (32.7%)			66 (13.0%) ***
χ^2^, *p* * 103.4, <0.001					

χ^2^, *p* = from the Chi square test comparing pupils and students’ sub-groups; *, **, *** *p* < 0.05, 0.01, and 0.001, respectively. Fisher exact test with Yates correction testing difference in frequencies in distinct categories. NS = not significant.The study groups were further divided according to the daily number of snack products consumed into the “low group” (once to several times per month), “medium group” (two to four times per week) and “high group” (once or several times per day). BMI did not differ between snack frequency subgroups, while high-school/university student ratio was significantly different. The majority of students (>50%) belonged to the “low group”, followed by almost one third of students in the “medium group”. Higher percent of university students belonged to the low intake group compared to the percent of high-school students, which suggested that younger students were more prone to frequently use salty snack products (Table 2).

**Table 2 nutrients-12-03290-t002:** Frequency of salty snack product consumption based on students’ age and weight status.

Parameter	Low Snack Intake *N* = 791	Medium Snack Intake *N* = 404	High Snack Intake *N* = 118	*p*
Age, years	21.5 ± 3.4	23.1 ± 2.1 ^aaa^	20.3 ± 2.7 ^aaa, bbb^	<0.001
BMI kg/m^2^	21.2 ± 2.7	20.8 ± 2.4	21.0 ± 2.9	0.157
<25 kg/m^2^	727 (59.7%)	384 (31.6%)	106 (8.7%)	χ^2^ = 5.4; 0.065
25–30	64 (66.7%)	20 (20.8%)	12 (12.5%)	
High-school students (*n* = 507)	276 (54.5%)	169 (33.4%)	61 (12.1%)	χ^2^ = 14.9; 0.001
University students (*n* = 806)	515 (63.8%)	235 (29.1%)	57 (7.1%)	

ANOVA test with post-hoc Tukey test is used for continuous variables comparison: age and BMI. ^aaa, bbb^
*p* < 0.001 vs. low snack and medium snack intake groups, respectively. Chi-square test is used for categorical variables. BMI, body mass index.

**Table 3 nutrients-12-03290-t003:** Logistic regression analysis of the variables (questionnaire’s items) that could predict a high level of salty snack product intake among the population of urban living students.

Population	All	High-School Students	University Students
	*p*	*p*	*p*
Gender m/f	0.943	0.401	0.537
Age (years)	<0.001	<0.01	0.378
BMI	0.515	0.464	0.834
Type of snack products	<0.001	<0.001	<0.001
Awareness of SP nutritive value	<0.05	0.540	<0.05
Morning consumption	<0.001	<0.001	<0.01
Midday consumption	<0.001	<0.001	<0.001
Evening consumption	<0.001	<0.001	<0.001
Motivation reasons	0.187	0.245	0.932
Number of main meals	0.117	0.271	0.436
Number of refreshments	<0.001	<0.001	<0.01
Self-perception of overall diet quality	<0.001	<0.001	0.182

*p*-values present significance level of distinct factors for high snack usage prediction; SP-salty snack product; m/f—male/female.

**Table 4 nutrients-12-03290-t004:** Salt content in snack products.

Product Type and Description	Salt Content (g/100 g)	G (g)	Percentage of Participants Reporting Using Certain Type of Snack Products (%)
Bakery products (crackers, fish-shaped products, *bake rolls*^®^ and *kubz*^®^, pretzels, sticks)	2.0–3.2 ^a^/2.6 ^b^	(18–160) ^c^/44.6 ^d^	31
Chips products (tortilla and potato)	1.3–1.6/1.5	(40–150)/46.9	20
Flips products (with peanuts, and classic)	1.4–2.4/1.9	(30–50)/23	7
*Clipsy* products^®^	1.7–3.5/2.6	45/45	7
Salted popcorn	1.5–1.9/1.7	(50–200)/90	20
Nuts and seeds	0.9–2.7/1.8	(42–200)/90	15

^a^ range for salt content values; ^b^ average value of salt content; ^c^ range value of the package size (g); ^d^ average value of the package size (g).

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
