# Peer review of "A Cross-Sectional Survey of Salty Snack Consumption among Serbian Urban-Living Students and Their Contribution to Salt Intake"

_nutrients, 2020, doi:10.3390/nu12113290_

Round 1

Reviewer 1 Report

The current study explores the salty snack consumption among high school and university students examining the cross-sectional behavioral patterns of students towards this snack category and relating those results to actual dietary values of salt included in the snacks consumed by the individuals. All this is performed in an attempt to relate the contribution of snacks to salt intake and associate this to possible health related long-term risks.

The study includes a satisfactory amount of sampling and the overall method applied for statistical analysis of results seems appropriate. The level of English language is satisfactory and the results are novel from the cross-sectional perspective (High school vs. University transition of students) and the zooming into salty snacks rather than examining the snack category (which is quite broad) in overall.

However, several major and minor issues with the paper lead me to suggest a major revision. These are related to several assumptions taken into the statistical approach, which the reviewer is not able to evaluate since the related material/ results are not present. The lack of presentation of important material for evaluation in the Materials and Methods section. The presentation of results and discussion points by the authors, which is quite unstructured, resulting to the confusion of the reader, obscuring the outcomes of the paper. In specific, in several instances results of minor importance may be underlined, while other of significant importance (as the authors state by the title “cross-sectional”) are hidden. Moreover, the authors need to become specific and avoid including in the paper vague statements about the results/ outcomes as well as discuss their results more actionably and within the perspectives of the paper (objective), rather than comparing them on the surface with existing literature that fits.

Within these frames, I will refrain myself at this point from making very specific comments and would rather point in the overall areas that need to be addressed and the actions that the authors need to undertake to improve the comprehension, underline the results and outcomes  of their work.

Consistency in reporting. It is very important for the reader to keep track of what you refer to in the manuscript text, so try to keep consistent on the phrasing or words used to define categories, it will significantly increase the comprehension. I give two examples below

In Line 18 of the abstract, you mention two groups that you examining: University students vs. High school students, as you mention. Please keep referring to them as mentioned above in the whole manuscript. The fact that suddenly mention them as students and pupils, or that you start exchanging University with faculty is very confusing for the reader and becomes impossible to follow.

Line 192-193: Low vs. high consumption groups; when you are reporting which subjects quota are included under each group be very clear and consistent which are the frequency categories. For example the “several times per week” and “once/ several times per month” mentioned in the text do not coincide with Table 2, the only categories “1-3 times per month”, “Once per week” “2-3 times per week” etc.

Abbreviations explain. Similarly to above, yet in the context of abbreviations. There are several instances where abbreviations are not explained at the first mention (Line 120 PASW); Please check. Moreover, be careful on how you introduce the abbreviations and the consistency that you use them thereafter. For example on Line 305 you introduce UL. However, you don’t introduce it correctly since UL stands for Upper Limit, which is something that is not mentioned in the text at all, while the text suggest that UL is something like a recommended daily intake. Please be specific on what you are referring to and cross check your text references to see if what you are referring to is being reflected in the text.

Questionnaire: In Lines 77-80; 83-86 you briefly describe the lifestyle and habit questionnaire used, which was created in specific for this survey. In my opinion, it is very important that you actually present the questions and available replies of the questionnaire, you mention it consisted of 12 items but which were those? What were the options for replies? Were they rated as continuous or non-continuous? This is all very important information. If the 12 questions are those included in Table 3 parameters, mention this and explain the available answers. In general it very hard to evaluate the data if you have no access to how the data were obtained. I would suggest that you include the questionnaire, the available answers, the type of variable and analysis undertaken as an Appendix

The Definition of snack. As you have mentioned in your article a snack can be perceived as multiple things (homemade snacks, ready to eat snacks etc.). Did you define what you meant by salty snack prior about snacking frequencies in your questionnaire? If yes, please insert this definition in the text of Material and Methods.

Exploratory factorial analysis, principal component analysis and factors chosen. You explain the statistical analysis for this in lines 131-137, mentioning that the aim was to classify the main factors, according to items nature and meaning. Then in lines 145-154 you mention model quality parameters, leaving only 4-5 lines to actually report on the results of the PCA, by briefly mentioned the 4 factors titles and explained variance. However, what specific items did these factors include? This will give an insight on why you grouped them as such. Moreover, where are these results used/ discussed? This feels like an analysis that is not used anywhere in the discussion, since after that the questions are analyzed individually. In Lines 159-161, you have a mention that read like a discussion/ specification of the environmental influence, however, this is very short and it should have been included in the discussion and not the results section. Very importantly, related to my mention of consistency above, you refer to these 4 dimensions as (extracted) factors (Lines 154-158). Later on (lines 200-201, Table 3) you refer to the individual questionnaire items as factors, this is very confusing to the reader. Please try to define and keep the same language from start to finish

Subgroup n=200 students undertaking DR1 & DR2. You mention that a sub-group of your total N of participants undertook the DR2 questionnaire, which were the participants used to calculate the salty snack exposure. It is very important here to mention a) were the DR1 results of the subgroup n=200 reflective of the overall results of the total N of participants b) what were the characteristics (sociodemographic, consumption group etc.) of the n=200 analyzed. Where they reflective of the population included in the overall study? Please elaborate and include these results in the text

Statistical part better explained (Lines 119-137). I would advise that you try to make the analysis part more clear. You don’t necessarily need to insert much more information, but rather use different paragraphs according to the theme and explain which data/ variables underwent each analyses and what type of data  that were, to ensure that the right types of test were used

Chi-square and “0” categories: Chi-square is not an appropriate test when the expected value of a category is below 5 (in that case an alternate test, e.g. Fisher’s exact test, should be used). However, in Table 1 you mention that you have analyzed the significance of “Frequency of snack consumption” with Chi-Square while the Never category is 0% and the several time per day is below 5%. Please re-analyze your results using an appropriate test

Table 1: The formatting of Table 1 does not follow the journals guidelines and is very confusing to read. Please make sure that the spacing is correct, the results are aligned and that different responses (Snack consumption in front of TV and computer) appear in different lines.

Results in Tables and Text. You repeat results (%) in the text while they are already included in the Tables, which is a duplication. I would suggest you avoid this. You can use this space instead to talk about the relative differences between your groups

High vs. low consumption groups. The two groups suggested, on which assumptions and further analyses are based, vary a lot not only in representation numbers (%), but also schematically, which creates issues with the interpretation of results later on. What were the criteria that actually drove you to consider as a low snacking frequency people that consume salty snacks to 2-3 per week? Do you expect that these people behave the same with people that may snack once per month? And please bear in mind that your categories are not covering all frequencies, a.k.a. if I eat snack all working days beyond weekend =5 days a week, what do I tick…? Daily or 2-3 times per week..? No real category is there for me. What I would suggest is that you reconsider your grouping. Maybe it will make sense if you actually created 3 groups, with a high, medium and low consumption of snacks. Please examine your results and consider what is more meaningful.

Table 2: The way you report significance is quite confusing. What is significant low vs. high? High school vs. University? The distributions of low vs. high between University and High school? Please make sure that you are clear which test is used for which analyses (don’t use “or” create a separate foot note) and align the P-values to the results you are referring to.

Discussion. In general, it is very hard to follow the discussion. The reader is lost into what type of results are discussed and in what context. What I can suggest is that you create sub-chapter addressing the discussion of results into context, according to the analysis perspectives that you have followed. For example, as a suggestion, you can create a subchapter addressing the results a) In overall b) Between High School and University students c) Between Low, Medium and High Frequency salty snack consumers (and it will be very interesting to see “who” were those, beyond their BMIs). This will allow you much better to discuss your results in connection to you aim : age related, demographic and behavioral differences, leading to higher consumption and if these factors vary among your cross sectional groups. Currently, a clear discussion is lacking and the parts of results underlined as important (a.k.a. discussed in plenum) seem random according to the literature found. The same applies for the results included in the abstract, since some of those where only slightly mentioned in the text. In specific, the only other mention of strategies suggested in the abstract (lines 28-31), as related to the current study is in line 328 of the discussion. However, the is no real discussion of how the outcomes of this study relate to the strategies suggested by the authors  

High School vs. University students Regression results in Appendix instead of discussion. While an important aspect of this study is the cross-sectional part, the results of the individual regressions per group are undermined by placing them in the appendix. I would definitely suggest that importance (P-values) of the Logistic regression analysis of factors of High school vs. University are placed in a table in the main manuscript. If you don’t want to overload the article with results you can create a table where similar to Table 1 you have ALL vs. High School vs. University students’ P-values for factors, and include the more detailed results about the model in the appendix. This especially important when one examines the regression analysis results of the individual groups. In detail, with the exception of the type of snack products, no other factor provides similar contributions to determining snack intake.  In specific for University students, seem as consuming snacks in the evening vs. high school students that consume them in the morning. Moreover, the motivation factor contributes to intake for University students, while the same does not apply for high school students. Whereas the opposite applies for the number of refreshments. Thereby, reporting the average results and discussing that motivation, number of refreshments etc. affect in general is not relevant, since the average model’s results are clearly related to one group and not both. This becomes more apparent when you look at Gender, Age, body weight and BMI (0.088). While these factors show a contribution in the average model, they do not contribute significantly in the individual ones, reflecting the different sampling (Males vs. Females distribution) between the two groups as well as possible physiological variations, related to body development from 16 to 25 years of age. These differences are worth of discussion, since they may depend greatly on the groups’ characteristics, routines, needs and way of life that vary and can help with aligning better strategies to reduce salt consumption from unhealthy snacks for each cross sectional population.

Author Response

Response to Reviewer 1 Comments

The current study explores the salty snack consumption among high school and university students examining the cross-sectional behavioral patterns of students towards this snack category and relating those results to actual dietary values of salt included in the snacks consumed by the individuals. All this is performed in an attempt to relate the contribution of snacks to salt intake and associate this to possible health related long-term risks.

The study includes a satisfactory amount of sampling and the overall method applied for statistical analysis of results seems appropriate. The level of English language is satisfactory and the results are novel from the cross-sectional perspective (High school vs. University transition of students) and the zooming into salty snacks rather than examining the snack category (which is quite broad) in overall.

However, several major and minor issues with the paper lead me to suggest a major revision. These are related to several assumptions taken into the statistical approach, which the reviewer is not able to evaluate since the related material/ results are not present. The lack of presentation of important material for evaluation in the Materials and Methods section. The presentation of results and discussion points by the authors, which is quite unstructured, resulting to the confusion of the reader, obscuring the outcomes of the paper. In specific, in several instances results of minor importance may be underlined, while other of significant importance (as the authors state by the title “cross-sectional”) are hidden. Moreover, the authors need to become specific and avoid including in the paper vague statements about the results/ outcomes as well as discuss their results more actionably and within the perspectives of the paper (objective), rather than comparing them on the surface with existing literature that fits.

Within these frames, I will refrain myself at this point from making very specific comments and would rather point in the overall areas that need to be addressed and the actions that the authors need to undertake to improve the comprehension, underline the results and outcomes  of their work.

Point 1: Consistency in reporting. It is very important for the reader to keep track of what you refer to in the manuscript text, so try to keep consistent on the phrasing or words used to define categories, it will significantly increase the comprehension. I give two examples below.

  1. In Line 18 of the abstract, you mention two groups that you examining: University students vs. High school students, as you mention. Please keep referring to them as mentioned above in the whole manuscript. The fact that suddenly mention them as students and pupils, or that you start exchanging University with faculty is very confusing for the reader and becomes impossible to follow.

Point 1 / A response:

We kept the expression university students and high-school students throughout the text (lines 106, 158-162, Table 1, 205-210, 214-216,  232-233, 241, 274-275, 307,  362, 372-273, 378-337, 430-43), as well as the expression “University” instead of “faculty” (lines 26)

  1. Line 192-193: Low vs. high consumption groups; when you are reporting which subjects quota are included under each group be very clear and consistent which are the frequency categories. For example the “several times per week” and “once/ several times per month” mentioned in the text do not coincide with Table 2, the only categories “1-3 times per month”, “Once per week” “2-3 times per week” etc.

Point 1 / B response: We accepted your suggestion (from point 11) to re-group our snack frequency categories and we divided subjects on 3 groups: low (1-3/month and 1/week), medium (2-4 times per week) and high users (1 or more times per day). All further analyses presented in the Tables 2 and 3 are performed on 3 above mentioned subgroups instead of 2 as in previous version of manuscript. In lines 223-224 these 3 groups are defined.

Point 2: Abbreviations explain. Similarly to above, yet in the context of abbreviations. There are several instances where abbreviations are not explained at the first mention (Line 120 PASW); Please check. Moreover, be careful on how you introduce the abbreviations and the consistency that you use them thereafter. For example on Line 305 you introduce UL. However, you don’t introduce it correctly since UL stands for Upper Limit, which is something that is not mentioned in the text at all, while the text suggest that UL is something like a recommended daily intake. Please be specific on what you are referring to and cross check your text references to see if what you are referring to is being reflected in the text.

Point 2 response:

Line 42: United States for abbreviation US;

Line 115-116: Association of Official Analytical Chemists for the abbreviation AOAC;

Line 134: Predicitive Analytics Software fot the abbreviation PASW;

Lines 408: Upper level (UL) was exchanged with recommended maximum level of intake (RMLI), the term that was used in cited WHO document.

Point 3: Questionnaire: In Lines 77-80; 83-86 you briefly describe the lifestyle and habit questionnaire used, which was created in specific for this survey. In my opinion, it is very important that you actually present the questions and available replies of the questionnaire, you mention it consisted of 12 items but which were those? What were the options for replies? Were they rated as continuous or non-continuous? This is all very important information. If the 12 questions are those included in Table 3 parameters, mention this and explain the available answers. In general it very hard to evaluate the data if you have no access to how the data were obtained. I would suggest that you include the questionnaire, the available answers, the type of variable and analysis undertaken as an Appendix.

Point 3 response:

Instead of giving all the questions in the Methods section of the Manuscript, a translation of the questionnaire is given in the Supplementary Data.

Point 4: The Definition of snack. As you have mentioned in your article a snack can be perceived as multiple things (homemade snacks, ready to eat snacks etc.). Did you define what you meant by salty snack prior about snacking frequencies in your questionnaire? If yes, please insert this definition in the text of Material and Methods.

Point 4 response: At the first part of the questionaire the examples of salty snack foods were given: chips, flips, popcorn, fried corn, salted sticks, pretzels, crackers, fish-shaped products, roasted and baked nuts, seeds, and salty peanuts.

In lines 286-288 deffinition of snacks was given.

Point 5: Exploratory factorial analysis, principal component analysis and factors chosen. You explain the statistical analysis for this in lines 131-137, mentioning that the aim was to classify the main factors, according to items nature and meaning. Then in lines 145-154 you mention model quality parameters, leaving only 4-5 lines to actually report on the results of the PCA, by briefly mentioned the 4 factors titles and explained variance. However, what specific items did these factors include? This will give an insight on why you grouped them as such. Moreover, where are these results used/ discussed? This feels like an analysis that is not used anywhere in the discussion, since after that the questions are analyzed individually. In Lines 159-161, you have a mention that read like a discussion/ specification of the environmental influence, however, this is very short and it should have been included in the discussion and not the results section. Very importantly, related to my mention of consistency above, you refer to these 4 dimensions as (extracted) factors (Lines 154-158). Later on (lines 200-201, Table 3) you refer to the individual questionnaire items as factors, this is very confusing to the reader. Please try to define and keep the same language from start to finish

Point 5 response:

In the Results section – part related to Exploratory factorial analysis we have added specific items which are a part of one of 4 factors in order to make this part clearer. The items are grouped into 4 factors by principal component analysis conducted by using SPSS statistical software. We prepared Table S2 for Supplementary file with all items grouped by Factors produced from PCA analysis. Also, we have added a comment in discussion section from this part of analysis.

Results section:

Lines 173-180: “The first factor was related to the main pattern of salty snack consumption (items: dominant place of consumption, frequency of consumption, time of consumption regarding the main meal, product type) and carried 35.2%; the second that described information influence on snack eating habits (awareness of nutritive value) carried 17,3%; the third 12.3% and was related to the general dietary habits (items: number of main meals, number of refreshments,  eating with non-alcoholic or alcoholic beverages, eating in front of TV); and the fourth was on personal preferences (faculty/high school,  motivational reasons for consumption,  day period related consumption) carried 9.9% of variance.“

Discussion:

Lines 287-302: “The products that were considered as salty snacks in this study were pre-packaged, available in markets, local shops or traffics near faculty/high-school, or from vending machines provided on school/university.

Reliability analysis regarding internal consistency, so as inter-item correlation showed very good validity of the FFQ used in this study. Multidimensionality of the questionnaire is analyzed by factor analysis (PCA analysis). Principal component analysis revealed several aspects of this questionnaire: pattern of salty snack consumption, influence of the label information on snack consumption, general dietary habits and personal preferences which are related to snack usage. The 4 main layers of the questionnaire enabled analyzing snack usage habits of Serbian urban-living students and to make some general conclusions in order to improve eating habits. This also could open the additional exploration possibilities of presented results in PCA extracted factors as separate variables, and their relationship with different study data, but this direction would go beyond the basic aims of the current investigation and is certainly going to be exploited in the future.“

Point 6: Subgroup n=200 students undertaking DR1 & DR2. You mention that a sub-group of your total N of participants undertook the DR2 questionnaire, which were the participants used to calculate the salty snack exposure. It is very important here to mention a) were the DR1 results of the subgroup n=200 reflective of the overall results of the total N of participants b) what were the characteristics (sociodemographic, consumption group etc.) of the n=200 analyzed. Where they reflective of the population included in the overall study? Please elaborate and include these results in the text.

Point 6 response: The correlation of 200 DR2 participants to 1313 DR1 participants was not tested, except that they were the same participants that filled in the lifestyle, food frequency questionnaires and DR1 2 months before. One sentence is added to the Method section to better explain the inclusion criteria of this sub-group of participants:

Lines 107-109: This second survey was not announced in advance and the inclusion criteria were previous engagement in the study and the availability to give answers to DR2 survey. Gender and age characteristics of this sub-group reflected the same characteristics of all participating students.

Point 7:

Statistical part better explained (Lines 119-137). I would advise that you try to make the analysis part more clear. You don’t necessarily need to insert much more information, but rather use different paragraphs according to the theme and explain which data/ variables underwent each analyses and what type of data  that were, to ensure that the right types of test were used.

Point 7 response:

We devided 2.5 Statistical analysis section (lines 134-155) in 3 paragraphs according to the theme to make it more visible which data underwent each analyses.

Point 8:

Chi-square and “0” categories: Chi-square is not an appropriate test when the expected value of a category is below 5 (in that case an alternate test, e.g. Fisher’s exact test, should be used). However, in Table 1 you mention that you have analyzed the significance of “Frequency of snack consumption” with Chi-Square while the Never category is 0% and the several time per day is below 5%. Please re-analyze your results using an appropriate test.

Point 8 response:

We changed the explanation of the used statistical test in paragraph 1 of the 2.5 Statistical Analyses section (lines 136-140) to make the text more clear about the use of Chi-square (C-S) test.

We added absolute numbers in the table in front of percent in order to make Table 1 clearer.

Point 9: Table 1: The formatting of Table 1 does not follow the journals guidelines and is very confusing to read. Please make sure that the spacing is correct, the results are aligned and that different responses (Snack consumption in front of TV and computer) appear in different lines.

Point 9 response:

We re-formatted Table 1 according to the journal’s guidelines.

We added one more category (Other) to the column “Place of usual snack consumption”. We added one more category (Other) to the column “Snack consumption time”.

We added absolute numbers in the table in front of percent in order to make a Table 1 clearer.

We put additional explanation to Table 1 footnote concerning the level significance and testing differences (lines 192-193).

Point 10: You repeat results (%) in the text while they are already included in the Tables, which is a duplication. I would suggest you avoid this. You can use this space instead to talk about the relative differences between your groups High vs. low consumption groups.

Point 10 response:

We eliminated duplicating percentages in the Results section (lines 207-208, 213-214, 217). We added new paragraph (lines 229-234) to clear the age differences between 3 consumption groups (see response to point 11).

Point 11: High vs. low consumption groups. The two groups suggested, on which assumptions and further analyses are based, vary a lot not only in representation numbers (%), but also schematically, which creates issues with the interpretation of results later on. What were the criteria that actually drove you to consider as a low snacking frequency people that consume salty snacks to 2-3 per week? Do you expect that these people behave the same with people that may snack once per month? And please bear in mind that your categories are not covering all frequencies, a.k.a. if I eat snack all working days beyond weekend =5 days a week, what do I tick…? Daily or 2-3 times per week..? No real category is there for me. What I would suggest is that you reconsider your grouping. Maybe it will make sense if you actually created 3 groups, with a high, medium and low consumption of snacks. Please examine your results and consider what is more meaningful.

Point 11 response:

Thank you for this suggestion. We tried to make results clearer and easier to comprehend. We accepted your suggestion to re-group our snack frequency categories and we divided subjects on 3 groups: low (1-3/month and 1/week), medium (2-4 times per week) and high users (1 or more times per day). All further analyses presented in the Tables 2 and 3 are performed on 3 above mentioned subgroups instead of 2 as in previous version of manuscript.

Point 12: Table 2: The way you report significance is quite confusing. What is significant low vs. high? High school vs. University? The distributions of low vs. high between University and High school? Please make sure that you are clear which test is used for which analyses (don’t use “or” create a separate foot note) and align the P-values to the results you are referring to.

Point 12 response:

Table 2.  Frequency of salty snack product consumption based on age and weight status is transformed according to your suggestion i.e. we have compared 3 groups: low, medium, high snack consumers in order to qualify. All continuous variables (age and BMI) we compared by ANOVA test with post-hoc Tukey test. All discrete variables (frequencies in different categories/sub-groups we have compared by using Chi-square test (number of subjects in different categories of BMI <25kg/m2 and 25-30 kg/m2; number of subjects in pupil and students groups). The type of statistical analysis is indicated below the table.

Point 13: Discussion. In general, it is very hard to follow the discussion. The reader is lost into what type of results are discussed and in what context. What I can suggest is that you create sub-chapter addressing the discussion of results into context, according to the analysis perspectives that you have followed. For example, as a suggestion, you can create a subchapter addressing the results a) In overall b) Between High School and University students c) Between Low, Medium and High Frequency salty snack consumers (and it will be very interesting to see “who” were those, beyond their BMIs). This will allow you much better to discuss your results in connection to you aim : age related, demographic and behavioral differences, leading to higher consumption and if these factors vary among your cross sectional groups. Currently, a clear discussion is lacking and the parts of results underlined as important (a.k.a. discussed in plenum) seem random according to the literature found.

Point 13 response:

We accepted the suggestion to create sub-sections with separate sub-titles: 4.1 Questionnaire reliability; 4.2 General characteristics of salty snack consumption patterns; 4.3 Factors associated with high frequency of salty snack usage; 4.4 Differences between High School and University students; 4.5 Contribution of salty snacks to daily salt intake; 4.6 Limitations and implications.

Several new aspects of results were pointed out, including snacking frequency between Low, Medium and High Frequency salty snack consumers (lines 229-234) and in 4.4 section of the Discussion.

Point 14:

The same applies for the results included in the abstract, since some of those where only slightly mentioned in the text. In specific, the only other mention of strategies suggested in the abstract (lines 28-31), as related to the current study is in line 328 of the discussion. However, the is no real discussion of how the outcomes of this study relate to the strategies suggested by the authors .

Point 14 response:

The end of the Abstract (lines 29-32) was changed because the strategies are possible implication and impact of this study, but were not its aim.

Point 15: High School vs. University students Regression results in Appendix instead of discussion. While an important aspect of this study is the cross-sectional part, the results of the individual regressions per group are undermined by placing them in the appendix. I would definitely suggest that importance (P-values) of the Logistic regression analysis of factors of High school vs. University are placed in a table in the main manuscript. If you don’t want to overload the article with results you can create a table where similar to Table 1 you have ALL vs. High School vs. University students’ P-values for factors, and include the more detailed results about the model in the appendix. This especially important when one examines the regression analysis results of the individual groups. In detail, with the exception of the type of snack products, no other factor provides similar contributions to determining snack intake.  In specific for University students, seem as consuming snacks in the evening vs. high school students that consume them in the morning. Moreover, the motivation factor contributes to intake for University students, while the same does not apply for high school students. Whereas the opposite applies for the number of refreshments. Thereby, reporting the average results and discussing that motivation, number of refreshments etc. affect in general is not relevant, since the average model’s results are clearly related to one group and not both. This becomes more apparent when you look at Gender, Age, body weight and BMI (0.088). While these factors show a contribution in the average model, they do not contribute significantly in the individual ones, reflecting the different sampling (Males vs. Females distribution) between the two groups as well as possible physiological variations, related to body development from 16 to 25 years of age. These differences are worth of discussion, since they may depend greatly on the groups’ characteristics, routines, needs and way of life that vary and can help with aligning better strategies to reduce salt consumption from unhealthy snacks for each cross sectional population.

Point 14 response:

Answer: According to your point 12 about inclusion of 3 groups low-medium-high snack consumers (instead two high-low in previous version of the Manuscript) we have again performed binary logistic regression (BLR) analysis comparing the two most opposite groups (low vs high) snack consumers in order to find what variables are significant predictors of high frequency snack consuming. Also, as you advised, we have added BLR analysis for high-school and university student’ groups separately. We also find that this analysis is very important for making valid conclusion about dietary habits of our young population. – both these changes are in detail presented in Supplementary Table 3.

In accordance with your suggestion we re-arranged Table 3 in the main text without overloading it, where we presented ALL vs. High School vs. University students’ P-values for factors determining snack intake.

The text in the Results section that explains Table 3 is also changed: lines 242-245 and 242-257.

The text in Discussion section that discuss new data from Table 3 is sub-section 4.3 (lines 342-367).

Reviewer 2 Report

In this study, the authors aimed to determine the patterns of salty snack consumption among Serbian urban-living students. The manuscript is so organized and very well-written. However, I have some minor comments.

  1. Page 1, line 17, please change “1.313” to “1,313” throughout the paper.
  2. Page 2, line 45, please add the reference.
  3. Page 2, line 61, please add the reference.
  4. Page 2, line 71, please describe the inclusion and exclusion criteria of this study. How many high schools and faculties were recruited into this study? Please describe your sampling method.
  5. Did you define snacks or salty snacks for students prior to data collection? For example, someone may consider salted peanut in his/her breakfast meal as a snack.
  6. Page 4, line 142, please add the unit for the age (year).
  7. Page 4, Table 1, please be consistent and place “often”, “sometimes”, and “never” in three different rows.
  8. Page 5, line 169, please change “food frequency questionnaire” to “FFQ.”
  9. Page 5, line 189, please be consistent and change “report” to “reported.”
  10. Page 6, Table 2, the authors stated that “Frequency of salty snack product consumption based on age and weight status” but they did not compare “low group” and “high group” in terms of age.
  11. Page 6, Table 3, please also add the parental educational level to the list of variables if you have data on that.
  12. Page 8, lines 247-249, “Surprisingly enough, the 247 “high snack intake group” had slightly lower BMI which do not support the hypothesis that higher 248 snacking frequency is associated with the obesity risk in adolescents and young people.” First, since you only collected data on salty snack consumption, this finding is not representative to all types of snacks. Second, the percentage of pupils in the “high snack intake group” is 51.6% compared to 37.3% in the “low snack intake group,“ which obviously lead to lower BMI in the “high snack intake group.”
  13. Page 9, lines 257-260, another possible explanation is that high snack intake increases the feeling of satiety which may lead to skipping main meals and lower energy intake.
  14. Page 9, line 275, there is no need to mention the statistical method used in the discussion.
  15. Page 9, lines 283-284, a recent study by Vatanparast et al. has investigated the time, location, and frequency of snack consumption among different age groups of Canadians (DOI: 10.1186/s12937-020-00600-5). It may be useful in terms of comparing the results for time and location of snack consumption among adolescents.
  16. Page 10, line 304, please remove “(The World Health 304 Organization)” since you previously defined WHO.
  17. Page 10, lines 330-332, please add the self-reported height and weight as a limitation to this study.

Author Response

Response to Reviewer 2 Comments

In this study, the authors aimed to determine the patterns of salty snack consumption among Serbian urban-living students. The manuscript is so organized and very well-written. However, I have some minor comments.

  1. Page 1, line 17, please change “1.313” to “1,313” throughout the paper.

We corrected the number: lines 17, 75, 157.

  1. Page 2, line 45, please add the reference.

In lines 45 and 47 two references are more clearly stated. The reference 6 is changed in the list of references to correspond the statement in the line 46.

  1. Page 2, line 61, please add the reference.

The new reference is added in line 63. All further references changed their number in the text and in the reference list.

  1. Page 2, line 71, please describe the inclusion and exclusion criteria of this study. How many high schools and faculties were recruited into this study? Please describe your sampling method.

In lines 76-83 more information was given on the type of schools/faculties that were participating in the study, as well as inclusion/exclusion criteria. The high-schools and faculties that where data collection was implemented were chosen according to their readiness to give consent to the study.

  1. Did you define snacks or salty snacks for students prior to data collection? For example, someone may consider salted peanut in his/her breakfast meal as a snack.

The salty snacks were defined in the questionnaire used for this study. The English translation of the questionnaire is in a Supplementary data.

  1. Page 4, line 142, please add the unit for the age (year).

The age unit (year) was added in line 160.

  1. Page 4, Table 1, please be consistent and place “often”, “sometimes”, and “never” in three different rows.

Table 1 is corrected according to the comment.

  1. Page 5, line 169, please change “food frequency questionnaire” to “FFQ.”

The correction is made in line 196.

  1. Page 5, line 189, please be consistent and change “report” to “reported.”

The correction is made in line 219.

  1. Page 6, Table 2, the authors stated that “Frequency of salty snack product consumption based on age and weight status” but they did not compare “low group” and “high group” in terms of age.

In Table 2 additional variable – age is added.

  1. Page 6, Table 3, please also add the parental educational level to the list of variables if you have data on that.

Unfortunately we do not have data on the parental education level.

  1. Page 8, lines 247-249, “Surprisingly enough, the 247 “high snack intake group” had slightly lower BMI which do not support the hypothesis that higher 248 snacking frequency is associated with the obesity risk in adolescents and young people.” First, since you only collected data on salty snack consumption, this finding is not representative to all types of snacks. Second, the percentage of pupils in the “high snack intake group” is 51.6% compared to 37.3% in the “low snack intake group,“ which obviously lead to lower BMI in the “high snack intake group.”

The sentence is changed in such a way to lessent the strenght of the statement – lines 321-323.

  1. Page 9, lines 257-260, another possible explanation is that high snack intake increases the feeling of satiety which may lead to skipping main meals and lower energy intake.

Thank you for the comment. We accepted and added it in lines 338-339.

  1. Page 9, line 275, there is no need to mention the statistical method used in the discussion.

In lines 343 the mention of the statistical method was eliminated.

  1. Page 9, lines 283-284, a recent study by Vatanparast et al. has investigated the time, location, and frequency of snack consumption among different age groups of Canadians (DOI: 10.1186/s12937-020-00600-5). It may be useful in terms of comparing the results for time and location of snack consumption among adolescents.

The suggested reference is used as a reference 21 and in lines 312-313 comparing with our results was done. All further references changed their number in the text and in the reference list.

  1. Page 10, line 304, please remove “(The World Health 304 Organization)” since you previously defined WHO.

The whole expression is removed and the abbreviation is kept – lines 402-403.

  1. Page 10, lines 330-332, please add the self-reported height and weight as a limitation to this study.

Self-reported height and weight are added in the limitations of the study - line 434.

Reviewer 3 Report

The authors investigated the association between salt consumption among Serbian urban-living students and contributing factors. The topic is interesting, but there are several critical drawbacks in this study, in my opinion.

  1. The detail inclusion criteria should be shown in the study population. There is no information when this study was conducted. The number of high schools and universities, and their locations should be shown.

  1. Approximately three quarters participants were female, indicating there should be some bias including participants.

  1. There is no description how DR2 participants were selected. I wonder whether all participants were informed of the second analysis. If they were informed, their activities would have been affected.

  1. Although the mean salt intake from salty snack is less than 10% of daily salt intake in Serbia, how do the authors think salt from salty snack affect their health or behaviors? Were there any results regarding their eating habits, for example, preference meat, or vegetable? In addition, do authors have any information in terms of exercise habits?

  1. In my opinion, the conclusion mainly relied on the logistic analysis. Therefore, the differentiation into the two groups, low snack group and high snack group, is very important. How did the author determine the standard? If the standard is changed, the conclusion should be different.

  1. The aim of study was not so clear. In addition, their messages from the results of this study was weak. These two points are very important.

Author Response

Response to Reviewer 3 Comments

The authors investigated the association between salt consumption among Serbian urban-living students and contributing factors. The topic is interesting, but there are several critical drawbacks in this study, in my opinion.

  1. The detail inclusion criteria should be shown in the study population. There is no information when this study was conducted. The number of high schools and universities, and their locations should be shown.

In lines 76-83 more information was given on the location, type of schools/faculties that were participating in the study, as well as inclusion/exclusion criteria for participants. The high-schools and faculties where data collection was implemented were chosen according to their readiness to give consent to the study.

  1. Approximately three quarters participants were female, indicating there should be some bias including participants.

This is the fact that we became aware of after the complition of data collecting process. This is stated as one of the limitations of the study (line 435).

  1. There is no description how DR2 participants were selected. I wonder whether all participants were informed of the second analysis. If they were informed, their activities would have been affected.

In lines 106-109 additional information on DR2 participants was given.

  1. Although the mean salt intake from salty snack is less than 10% of daily salt intake in Serbia, how do the authors think salt from salty snack affect their health or behaviors? Were there any results regarding their eating habits, for example, preference meat, or vegetable? In addition, do authors have any information in terms of exercise habits?

We did not collect data on the exercise level or food preferences in the questionnaire.  The potential effect of salty snack on daily intake is in the range of dairy products which are one of the main food categories and the authors find this important from the potential health effects - that was expressed in lines 419-421.

  1. In my opinion, the conclusion mainly relied on the logistic analysis. Therefore, the differentiation into the two groups, low snack group and high snack group, is very important. How did the author determine the standard? If the standard is changed, the conclusion should be different.

We changed the categorization in Table 2. We included the third, „medium“ group of users as a new category. This change affected the statistics and new results are presented in Tables 2 and 3. We think that this categoization is more relevant to the wide range of using frequencies that were found among the participants.

  1. The aim of study was not so clear. In addition, their messages from the results of this study was weak. These two points are very important.

We added a sentence in the Introduction section to strenghten the aim of the study – lines 70-71.

In lines 440-441 we have added new messages that we found important  as results of this study.

Round 2

Reviewer 1 Report

The reviewer would like to comment the authors for the extensive work they put into, replying to the reviewer's comments and addressing the corresponding changes in the text. The manuscript has improved in a great extent and is a very easy and interesting read. 

I have no further comments, just a few formatting suggestions that the authors may like to address

Table 3: needs formatting according to journals guidelines

There are several instances that faculty is still used instead of University or there is not specification who are the students the sentence refers to, fw examples: Lines (track change version):205-206, 267-268, 444 etc.

Sentence lines 519-523 (track change version), is quite long, think of breaking it down to increase readability 

Reviewer 3 Report

Although manuscript has improved, there are a few critical concerns in this manuscript.

1.I think the following sentence has a critical contraction.

line 102-103

“This second survey was not announced in advance and the inclusion criteria were previous engagement in the study and the availability to give answers to DR2 survey.”

If the author asked the availability of DR2 survey, the participants should have been aware of the second survey, which should have affected the results.

In addition, I do not think the number of students who could participate in DR2 survey was 100 each in university students and high school student.

Consequently, there should be some selections regarding DR2. I think the authors did not reply properly how they selected the participants in DR2.  

  1. I previously asked how the author divided the category for the logistic regression analysis. I did not want the authors to divide the participants further into the three class.

Since the author analyzed the factors which affected a high level of salty snack products, adding medium snack intake does not make sense, which would not affect the results regarding table 3 because the analyses were focused on the high salt intake group. However, the results have changed from the previous table 3 to the present table 3, including supplement table 3. For example, in the previous version the gender and age were associated with the high level of salty snack products, however, these significances were not observed in the revised version.

I cannot understand these changes in the results. If the results have completely changed different by modifying the categories, I cannot believe the main result of this study. This is the very fact which I meant the previous review.
